# Suffering out of sight but not out of mind – interpreting experiences of sick leave due to chronic pain in a community setting: a qualitative study

Åse Lundin ![ORCID],[1,2] Inger Ekman,[1,2,3] Sara Wallström ![ORCID],[1,2,4] Paulin Andréll,[5,6] Mari Lundberg[2,7]

For numbered affiliations see end of article.

**Correspondence to**
Åse Lundin; ase.lundin@gu.se

## ABSTRACT

**Objective** Chronic pain is a complex health problem affecting about one-fifth of the European population. It is a leading cause of years lived with disability worldwide, with serious personal, relational and socioeconomic consequences. Chronic pain and sick leave adversely affect health and quality of life. Thus, understanding this phenomenon is essential for reducing suffering, understanding the need for support and promoting a rapid return to work and an active lifestyle. This study aimed to describe and interpret persons' experiences of being on sick leave due to chronic pain.

**Design** A qualitative study with semistructured interviews analysed using a phenomenological hermeneutic approach.

**Setting** Participants were recruited from a community setting in Sweden.

**Participants** Fourteen participants (12 women) with experiences of part-time or full-time sick leave from work due to chronic pain were included in the study.

**Results** Suffering out of sight but not out of mind was the main theme of the qualitative analysis. This theme implies that the participants' constant suffering was invisible to others, causing them to feel they were not being justly treated in society. Feeling overlooked led to a continuous struggle for recognition. Moreover, the participants' identities and their trust in themselves and their bodies were challenged. However, our study also revealed a nuanced understanding of the experiences of sick leave as a consequence of chronic pain, where the participants learnt important lessons, including coping strategies and re-evaluated priorities.

**Conclusions** Being on sick leave due to chronic pain threatens a person's integrity and leads to substantial suffering. An enhanced understanding of the meaning of sick leave due to chronic pain provides important considerations for their care and support. This study highlights the importance of feeling acknowledged and being met with justice in encounters with others.

## INTRODUCTION

Chronic pain has reached alarming levels, particularly among older adults, and is one of the leading causes of years lived with disability.[1] Pain of moderate-to-severe intensity affects

## STRENGTHS AND LIMITATIONS OF THIS STUDY

⇒ Qualitative research is important in enlightening the personal experiences of pain and its consequences, uncovering information not accessible through other methods.

⇒ Exploring the meanings of lived experiences can help us understand and improve our practices in healthcare.

⇒ Studies using interpretative qualitative methods are encouraged within the field of chronic pain and address a knowledge gap in the existing literature.

⇒ In this study, only 2 out of the 14 participants were men, yielding a distribution that might have affected our findings.

19% of the European population.[2] The socioeconomic burden of chronic pain is substantial, mainly for working-age people, with the bulk of indirect costs related to absence from the workforce.[3] In the USA, the total cost associated with chronic pain in adults ranges from US$560 to US$635 billion, exceeding the annual costs estimated for heart disease (US$309 billion), cancer (US$243 billion) and diabetes (US$188 billion).[4] In Sweden, the cost of chronic pain diagnoses was estimated to be €32 billion annually.[3] Chronic pain has a significant impact on quality of life and is closely associated with depression,[5] increased suicide rates,[6] premature mortality and reduced life expectancy.[7 8] Because a key feature of pain is that it is a 'personal experience that is influenced to varying degrees by biological, psychological and social factors',[9] a person's experience or report of pain is of particular interest and should be taken seriously.[10] Qualitative pain research has been increasing in recent years, exposing aspects of pain experience that is not accessible through other approaches.[11] Previous studies of people's experiences with chronic pain revealed feelings of restriction,

loss and limitation of daily life,[12 13] loneliness and silent suffering.[14] These studies also showed that chronic pain had a pronounced effect on relationships, family life and work.[13] In addition, being on sick leave negatively affects a person's life. Research on the experiences of work absenteeism because of illness has been done in different populations and for different health conditions, revealing negative experiences including loss of identity,[15] social exclusion and a struggle to adapt to reduced work capacity.[16] Being on sick leave also led to feelings of uncertainty and insecurity,[17] stigma, helplessness and a loss of independence.[18]

Despite implementing evidence-based practices (eg, interdisciplinary rehabilitation), many persons with chronic pain do not return to work.[19] Qualitative pain research has often focused on the barriers and facilitators people with chronic pain face when returning to work.[20 21] However, there is limited research on the personal experiences of being on sick leave due to chronic pain. Some studies reflect on immigrant women's experiences of long-term sick leave due to chronic pain. These studies found that the women felt excluded and rejected,[22] struggling with an invisible physical ailment.[23] Other studies have explored the phenomenon on a descriptive level.[24] Yet, researchers have been encouraged to go beyond description and towards interpretative engagement while performing qualitative analyses of pain.[11] In line with this approach, we sought to understand the essential meaning of sick leave due to chronic pain. A deepened understanding of the personal experiences of the phenomenon is vital to reducing suffering, better understanding their needs and facilitating a return to working life.

This study aimed to describe and interpret persons' experiences of sick leave due to chronic pain.

## METHODS

The authors followed the Standards for Reporting Qualitative Research guidelines[25] to ensure the trustworthiness of the findings.

### Design

In this study, we chose a qualitative interview design to explore participants' experiences of living with chronic pain and on sick leave. Data (transcripts of semistructured interviews) were analysed using phenomenological hermeneutics inspired by Ricoeur[26] and developed by Lindseth and Norberg.[27] The method has been used in numerous health and care studies.[28–30] The method aims to understand the meaning of the participants' lived experiences by interpreting their narratives.

### Participants and setting

The inclusion criteria for this study were: having experience of part-time or full-time sick leave attributable to chronic pain (duration of pain >3 months), speaking Swedish or English and being able and willing to participate in an interview. The recommended target size in qualitative interview studies varies, but phenomenological research often involves 5–25 participants who have experienced the phenomenon in question.[31] During the interviews, three of the present authors (ÅL, SW and IE) regularly evaluated the richness of the collected data relating to the research aim to help determine when sufficient participants had been included.

The participants were recruited through patient organisations and associations focusing on pain or pain-related conditions. A request for participation was sent through their email, newsletter or Facebook pages. The participants could choose the interview location and whether they wanted to meet face-to-face or through digital communication.

### Data collection

Data were collected via individual semistructured in-depth interviews that were audiorecorded and transcribed verbatim. All interviews and transcripts were confidential using numerical coding, and precautions were taken to safeguard the identity of the participants.

An interview guide was used (see online supplemental material). All interviews opened with the question, 'I understand you have been living in pain for a long time. Would you tell me what this time has been like for you?.' Follow-up and probing questions such as 'Can you give me an example?' and 'How did that make you feel?' encouraged further narration. The interview guide was not used as a strict list of questions, but to help propel the conversation forward and ensure the research questions were covered during the interview. Some participants freely narrated, while others needed more guidance to address the research questions.

### Analysis and interpretation of data

Phenomenological hermeneutics comprises three interrelated parts: naïve reading, structural analysis and interpretation of the whole.[27] During the naïve reading, the text was read repeatedly to grasp its meaning as a whole. In the structural analysis, the text related to the aim of the study was divided into meaning units that were then abstracted and formed into subthemes, themes and a main theme. Qualitative data analysis software (QSR NVivo V.R1) was used in the structural analysis. The analysis was a continuous process moving between the parts and meaning units of the text and the overall understanding. Three authors (ÅL, ML and IE) met regularly during this process to discuss the collected data and the emergence of subthemes, themes and the main theme to highlight different perspectives, add breadth and nuances, and ensure a congruent reflection and understanding of the narratives. All parts of the text relevant to the research question were included in the analysis. During the final part of the analysis, the text was reread. The initial naïve reading, the structural analysis and the authors' preunderstandings were combined to interpret the overall meaning of the text. We used research related

to the area, theoretical literature and discussions among the authors to gain a better understanding of the text.

## Research characteristics and reflexivity
ÅL is a physiotherapist with several years of clinical experience in primary healthcare. The author has frequently encountered patients seeking care for chronic pain who cannot work because of their condition. ÅL has previously done qualitative research but not in areas related to pain. Of the coauthors, two have extensive experience in research on pain (ML and PA) and three have extensive experience working with qualitative analyses (IE, ML and SW).

## Ethical considerations
Written consent was obtained from each participant. The study was approved by the Swedish Ethical Review Authority DNR 2020-02491 and complied with the principles in the Declaration of Helsinki.[32]

## Patient and public involvement
Before the study, several patient associations were contacted to help recruit participants. These patient organisations and their channels will also disseminate the study results to reach a wider audience beyond the academic community.

## FINDINGS
Of the 18 people who contacted the first author (ÅL) regarding participation, 3 decided not to go through with an interview and 1 did not attend the booked interview. Fourteen participants (12 women) were included in the study and were interviewed. Participant characteristics are listed in table 1.

| Table 1 | Participant characteristics |
|---|---|
| **Characteristics** | **(n=14)** |
| Men/women | 2/12 |
| Mean age, years (range) | 49 (23–80) |
| Current work status | |
| Full-time sick leave | 4 |
| Part-time sick leave | 5 |
| Returned to work | 4 |
| Retired | 1 |
| Highest completed education | |
| Primary school (9 years) | 2 |
| High school | 3 |
| Other upper secondary education | 4 |
| University degree | 5 |
| Duration of pain | |
| 1–3 years | 2 |
| 4–9 years | 1 |
| ≥10 years | 11 |

The location and causes of chronic pain varied among the participants. Low back pain, neck pain, rheumatism, fibromyalgia, shoulder pain, arthritis, endometriosis, joint pain due to Ehlers-Danlos syndrome, whiplash injury and spinal injury were represented among the participants.

The interviews, lasting 41–90 min, were conducted in 2021 by the first author. Of the 14 interviews, 10 were performed through digital channels, 2 in the participants' homes and 2 on university premises.

## Naïve reading
The naïve reading of the text indicated that living with chronic pain and being on sick leave significantly impacted the participants' lives and how they identified themselves. The feelings of exclusion, loss and unfamiliarity prevailed throughout the naïve understanding. The participants struggled to manage a condition experienced as invisible to others, hindering participation in daily activities and work participation.

## Structural analysis
The structural analyses partly validated our first impressions. These analyses also helped nuance and emphasise other aspects of the phenomenon, mainly regarding dependency and relations with institutions, understandings of the phenomenon that went unnoticed in the first part of the analysis.

An overview of the main theme, themes and subthemes from the structural analysis is given in table 2 and further described below.

## An identity under threat
Being on sick leave because of chronic pain affected and threatened the participants' views of themselves and their identity, significantly influencing their relationships with others. This theme was expressed in three subthemes: Becoming dependent, Being excluded and Feeling flawed.

### Becoming dependent
The participants felt they no longer recognise themselves concerning their capabilities. They had previously viewed themselves as independent and able to care for and fend for themselves. However, being on sick leave due to chronic pain made them increasingly dependent on others. Becoming dependent involves relying on help from family and friends, outsourcing physical household chores, and becoming economically reliant on their partners or others to pay for their dwelling or provide for the children. The participants experienced frustration from relying on the benefits and goodwill of authorities concerning their work status and sick leave payments. Decisions were often out of the participants' control, causing distress and feelings of worthlessness and powerlessness. One participant who was denied access to sick leave benefits said,

> Ultimately, you feel like you are just a burden to your husband [crying]. I have no income to get by myself.

| Table 2 | Overview of the main theme, themes and subthemes | |
|---|---|---|
| **Suffering out of sight but not out of mind** | | |
| **An identity under threat** | **Trapped in a foreign body** | **Navigating uncharted waters** |
| Becoming dependent<br>Being excluded<br>Feeling flawed | Losing bodily trust<br>Feeling constantly tired | Constantly chasing recognition<br>A journey towards unexpected lessons |

I cannot earn a living. And if he cannot provide service, help me, or fix the car, we have to sell it because I can't do it. #4

### Being excluded

The presence of pain and being on sick leave led to feelings of exclusion from the workplace and losing value as an employee and person. The participants had previously taken pride in being willing to work hard. They wanted to return to work and be a part of the social life and the routines they had at the workplace that made them feel meaningful and appreciated. Having to go on sick leave was experienced as imprisonment or punishment and feeling left behind.

> Being at home it…I think it is the worst punishment. The days…the days are long when you are healthy so you can imagine when you are sick and restricted and…and being alone in the house and my husband is working and my daughter is at school and then you think a lot of negative thoughts and you have to occupy the brain, I guess. #14

### Feeling flawed

Because the participants lived with pain and could not work, they felt flawed as persons, parents and participants in daily life. They struggled with a loss of engagement, unhappiness, increased anger, a short temper towards family members and other personality changes. They often had to decline participation in social activities or hobbies against their wishes. It affected their relationships with family and friends, creating grief, largely concerning being unable to be there for their friends, spouse or children as they used to be.

> But you still feel that it's like you are not sufficiently there. So, you also have a constant feeling of guilt that "no, mommy isn't able to…". #6

### Trapped in a foreign body

Suffering from chronic pain meant not recognising the body as the familiar place it used to be, creating conflicts that made it difficult to cope with daily and working life. This theme was expressed in two subthemes: Losing bodily trust and Feeling constantly tired.

### Losing bodily trust

Experiencing distressing physical symptoms combined with intensive and unreliable pain fluctuations was tiresome for the participants. Not knowing how their body would function or react daily created a loss of trust and a feeling that their body was not the safe and trusted home it once was. One participant described it as always wearing a heavy diving suit, interpreted as a conflict between what the participant wanted and what the body could do. Another participant, who worked as a caretaker, described how the rapid increase in pain intensity made her feel insecure and lose trust in her body while helping and interacting with the elderly in their homes.

> Then I would make supper and help them into their nightclothes, and then you ought to…some of them who were in wheelchairs would have toilet visits and then do many things. And I dropped pots and cups on sinks, and they made a bang and these older people, 80–90 years old, were terrified! And I got so much pain that I could barely lift this pot back up again, or the cup or… I mean, I would have hot, both hot water and stuff. I'm terrified that I would drop an old…/…/And then I did not want to go back there when I know I do not, that I cannot hold. How can I hold a whole person when I cannot even grab a cup? #4

### Feeling constantly tired

The participants experienced that fatigue and the inability to sleep were a greater disturbance in their daily lives and their ability to work than the pain itself. The little energy possibly gained through the night was quickly drained the next day when coping with the fatigue associated with the pain. Sleepless nights, the side effects of sleep medication and not feeling rested were associated with stress and suffering. Being on sick leave and not having to get up for work opened up the possibility of resting a little during the day instead of the night, which was interpreted as a small window of opportunity to find relief.

> Because it's, it's, it's actually the worst, really. The pain, you can do something about it, you can, it can be eased…but the fatigue cannot be relieved in the same way. #10

### Navigating uncharted waters

Living in pain and not being able to participate in work was interpreted as navigating through the unfamiliar, a route filled with unpleasantness and uncertainties (but sometimes helpful surprises). This theme was expressed in two subthemes: Constantly chasing recognition and A journey towards unexpected lessons.

## Constantly chasing recognition

Living with pain meant confronting mistrust or suspicions from different parts of society. The pain was very tangible for the participants, but they experienced their condition as hidden from the outside world. This perceived invisibility made the participants feel invariably questioned, especially by the authorities, the healthcare system and the workplace. They felt that they were seen as complainers, lazy and making up or exaggerating their symptoms, which led to an endless pursuit to be believed and have their suffering recognised and taken seriously. One participant described not being met with support from the healthcare system early enough and how that might have prolonged her sick leave.

> If the scenario had been such that the healthcare system took care of me and helped me, I wouldn't have had… even though I will always have problems because I'm chronically ill, I will always have problems, but it hadn't been that noticeable. It would not have been so extensive and so often and so strong, which would have meant that I could have been able to handle it and I would have been able to work #9

Instead, the participants continuously fought to receive a diagnosis that could be shown through objective assessment to explain their pain and validate their getting access to treatment and their leave of absence from work.

> It does not matter what I say because it feels like they see my smile and that I am active and walking that "well, but then she is not so ill." And if I had gone back to work, "but then you are not so ill," and it starts all over again!/…/Just because it is not visible on a damn test…That you couldn't just, you know, you can connect cars to computers, couldn't you get a frigging body scan and *swoosh* where it lights up red, she's in pain there! #3

## A journey towards unexpected lessons

The participants were forced to find different coping strategies and new routines because of the unrelenting fatigue and constant pain. This endeavour was interpreted as a bumpy journey with uncertainties and worries, especially regarding their future and work situation if their condition deteriorates. The participants' experiences of living with pain were not all undesirable, as some had learnt meaningful lessons along the way. They gained new insights into the importance of not pushing oneself too hard, that rest is essential and that it is not beneficial to let work dominate life. They identified abilities and resources within themselves that they had not previously noticed that helped them cope with their new situation. They also re-evaluated their priorities regarding relationships and family and the importance of appreciating the small things in life.

> I think I'm better now at thinking that the small things are good and pleasant. It may be enough that I go for a walk in the woods and look at the water. You will still be all right, or better. In a way that I didn't …I didn't see the qualities of the little things in life quite the same way. Maybe it was the bigger things that were more important. The rest was mostly just everyday things. I do not think that way anymore; instead, I believe that there are small things that are important after all. #8

## Interpretation of the whole

The naïve reading, the structural analysis and the authors' preunderstandings were integrated to interpret the overall meaning of the text material. The interpretation was inspired by Ricoeur's philosophy and ethical intentions of 'aiming at a good life with and for others in just institutions',[33] emphasising that ethics is prioritised over morality or norms. Ricoeur's philosophy on just institutions is of particular interest to the current study as it is essential that people feel justly treated in their contact with institutions[34] such as the healthcare system, the authorities or the workplace. With pain often experienced as an invisible illness to others, there was a continual pursuit to have their life situation validated and recognised. We interpret this as being about justice, a feeling that they were not receiving or being met with the justice they deserved. During their sick leave, the participants' identities were threatened by feelings of flaws, exclusion and dependency. They felt trapped in their bodies, which they experienced as foreign and thus not their own. The themes were interpreted as being connected by processes invisible to others, including institutions, but still brought much suffering to the participants. This suffering was exacerbated by the feelings of not being treated fairly. Although the participants' struggles were hidden from the outside world, they were real to the participants, affecting all aspects of life. Based on this, the meaning of living with and being on sick leave due to chronic pain was interpreted as suffering out of sight but not out of mind, which is the main theme. It is a play on the idiom 'out of sight, out of mind', meaning that you soon forget people (or things) that are no longer visible or present. For the participants, however, the pain and suffering were permanent and real.

## DISCUSSION

The main finding of this study is that the participants experienced their suffering as invisible to others, causing them to feel that they were not being justly treated in society and invariably fighting for recognition of their condition. Moreover, being on sick leave due to chronic pain challenged the participants' identities, leading to a loss of trust in themselves and their bodies. Contrary to previous research, which has mainly underlined the hardships and adversities of sick leave,[16 17 22 23] our study revealed a nuanced understanding of the experiences of sick leave due to chronic pain. Despite their difficulties,

and sometimes to their surprise, the participants felt that this situation had taught them new things about themselves, forced them to develop new coping strategies, and allowed them to embrace other aspects of life. Research has shown that supportive experiences (eg, meeting respectful healthcare professionals) are important to those on sick leave[18] and that possessing problem-solving coping strategies significantly reduces sick leave.[35] In our study, we also interpreted the participants' learning experiences as something that stretched over a broader context and that adds value to relationships, family life and a positive approach to life. Our findings are important because if the participants had gained these skills and coping strategies earlier, their sick leave may possibly have been shortened or prevented. The participants in our study also explained that they might have continued working if their struggles had been recognised at the workplace or by the healthcare system from the start. We interpret this inadequacy as lacking the institutional support the participants needed.

The experience of being in a foreign body when suffering from poor health has been described in other studies. In illness, the body can be perceived as 'unhomelike' and as a 'broken tool' no longer in control, resulting in a loss of command over one's own body.[36] Suffering from chronic pain could also be interpreted as experiencing the body as an obstruction and being confronted with having a body that feels different from when healthy.[37] Our participants noted how they were affected by several neurophysiological changes due to pain, including fatigue, stress and mood changes (eg, increased anger). According to our clinical experiences, this might represent a lack of knowledge of the dialectical relationship between the body and psychological or behavioural changes. Increasing the understanding of the relationship between the mind and body and how this interaction affects the pain experience is an integral part of pain rehabilitation. Pain neuroscience education has been shown to increase this knowledge and reduce disability among patients suffering from chronic pain.[38] Our study also found that the losing familiarity with the body when suffering from pain hindered the participants' return to work. The ability to return to work was not only a question of the physical strength needed to perform certain work tasks, but about trusting their capabilities and feeling safe in their bodies in the workplace setting. As shown in previous research, the experience of moving while in persistent pain implies more than just physical restriction. It challenges and threatens a person's lifeworld and existence.[39] In the words of Merleau-Ponty, our lived body is our access to or 'a vehicle of being in the world'.[40] When our body changes (ie, when trapped in an unknown body due to pain), it alters how the world is perceived.

Our participants viewed their identities as challenged or under threat while on sick leave owing to chronic pain. This feeling is consistent with previous findings on the sick leave process due to other health conditions, expressed as being reduced as a person and not recognising oneself[15] or experiencing a changed self-perception.[18] Our participants depicted a strong connection between identity and work. Losing contact with the workplace meant forfeiting who they were as productive elements of society. The authenticity of pain conditions as a cause of work absence has been questioned by some employers, suggesting that workers are 'lazy'.[21] In contrast, our study shows that the participants had a strong desire to return to work and experience the benefits that were linked to being part of the workforce.

We interpreted the experiences of constantly chasing recognition while facing distrust and suspicion from others as not being met by just institutions. A just institution means that each person, when in contact with, for example, the healthcare system senses justice on a fundamental level, that is, is treated well and considered an equal partner in the care process.[34] A just institution, a crucial part of person-centred care, allows persons to express their needs and abilities and can even offer a sense of partnership.[34] However, our participants reported feeling subservient and helpless, with their lives dependent on the goodwill of institutional gatekeepers. Previous research on sick leave has observed similar experiences, that is, instead of a partnership, the rules and regulations make the stakeholders become counterparts, leaving little flexibility for the person.[15] But being on sick leave due to chronic pain was also interpreted as different from other health issues, mainly because of its invisibility and the lack of objective assessment of pain symptoms. A social stigma is attached to chronic pain.[41] In our study, one participant wondered whether the workplace or authorities would have looked at her differently if she had cancer rather than a chronic pain condition. In the 2018 International Classification of Diseases 11th revision (ICD-11), a coding system for chronic pain was included for the first time.[42] Acknowledging and recognising chronic pain as a disease in itself and not just a symptom is an opportunity to improve treatment and our understanding of chronic pain and its management.[42] Another benefit is that this new classification may potentially shift the focus from finding a cause to managing the impact of the chronic pain for the patient (eg, through multimodal treatments).[42] The participants in this study often fought hard to receive a diagnosis and to be able to objectively prove that they had pain to the healthcare professionals in order to receive the right help. The new ICD-11 can hopefully be part of acknowledging the importance of understanding in rehabilitation and in the process of returning to work, as well as potentially reducing stigma by increasing the legitimacy of chronic pain.

### Methodological considerations

In the current study, only 2 of the 14 participants were men. This skewed distribution might partly be explained by research showing that women report a higher prevalence of chronic pain[43] but could also be influenced by other factors (eg, women being over-represented or more

active in the patient organisations used to recruit participants). The study also explored the 14 participants' narratives, and their experiences might not apply to the experiences of other persons on sick leave as a result of chronic pain. However, qualitative research does not argue that the findings are universal and generalisable; rather, it explores possible meanings of lived experiences to understand and improve our practices in healthcare.[27]

Sample size in qualitative studies is often a subject of discussion, where no power calculations can determine the sample size needed.[44] In phenomenological research, sample sizes of between 5 and 25 persons are commonly suggested.[31] But determining sample size and when no further sampling is needed is a matter of judgement and experience in evaluating the quality of already collected data to the aims and methods used.[44] During the inclusion process and inspection of the interview data, the material was discussed within the research group and deemed to contain rich and varied narratives of the participants' experiences. Thus, the interview data sufficed to answer the research question using the phenomenological hermeneutic method. Given the richness of the interview data, further analyses that focus on other aspects of the collected data are planned.

Another potential limitation is related to the COVID-19 pandemic restrictions. Because of the restrictions, most interviews were performed digitally, which could have resulted in differences in the interactions and dynamics during the interviews compared with face-to-face encounters. A recent review showed that digital interviews could adversely affect the interviews, including experiencing technical issues or difficulties reading visual cues or body language. Yet, the review also found that richer data could be extracted through a digital approach compared with face-to-face interviews, when investigating sensitive topics (eg, health issues), most likely because the participants felt more in control of the situation and the setting and thus felt more relaxed.[45]

## CONCLUSIONS

This study underscores the experiences of persons on sick leave due to chronic pain, where feelings of being invisible and overlooked often dominate. They do not feel that they are met with justice in their encounters with their surroundings which creates existential concerns and threatens their identities, bodies and interactions with others. In contrast to previous research, which has mainly spotlighted the adversities of being in the sick leave process, our study revealed a nuanced understanding of the participants' experiences of sick leave due to chronic pain. A deepened understanding of this phenomenon provides important considerations for the care and support of these persons, as well as means for joint planning of rehabilitation towards a return to work. It is important to understand their lifeworld to acknowledge and raise the issues of existential threat, stigma, invisibility and feelings of injustice that these persons

experience in encounters with others. Suggestions for future research include further investigating what factors might facilitate a return to work after being on sick leave due to chronic pain and exploring the impact of the new ICD-11 classification on the personal experiences of living with chronic pain.

**Author affiliations**
[1]Institute of Health and Care Sciences, Sahlgrenska Academy, University of Gothenburg, Gothenburg, Sweden
[2]University of Gothenburg Centre for Person-Centred Care (GPCC), Sahlgrenska Academy, University of Gothenburg, Gothenburg, Sweden
[3]Department of Medicine, Geriatrics and Emergency Medicine, Sahlgrenska University Hospital/Östra, Gothenburg, Region Västra Götaland, Sweden
[4]Department of Forensic Psychiatry, Sahlgrenska University Hospital, Gothenburg, Region Västra Götaland, Sweden
[5]Department of Anaesthesiology and Intensive Care Medicine/Pain Centre, Sahlgrenska University Hospital/Östra, Gothenburg, Region Västra Götaland, Sweden
[6]Department of Anaesthesiology and Intensive Care Medicine, Institute of Clinical Sciences, Sahlgrenska Academy, University of Gothenburg, Gothenburg, Sweden
[7]Department of Health Promoting Science, Sophiahemmet University, Stockholm, Sweden

**Acknowledgements** We acknowledge the participants for taking the time and effort to share their narratives and experiences.

**Contributors** All authors participated in the conception and design of the study. ÅL recruited participants, conducted and transcribed the interviews, and performed the analysis and interpretation of data with critical input from IE and ML. The manuscript was drafted by ÅL and critically revised and edited by IE, ML, SW and PA. All authors read and approved the final version of the manuscript and agree to be accountable for all aspects of the work in ensuring that questions related to the accuracy or integrity of any part of the work are appropriately investigated and resolved. ML is responsible for the overall content as guarantor.

**Funding** This work was supported by FORTE grant number 2019-00718 and the University of Gothenburg Centre for Person-Centred Care (GPCC), grant number N/A.

**Competing interests** None declared.

**Patient and public involvement** Patients and/or the public were involved in the design, or conduct, or reporting, or dissemination plans of this research. Refer to the Methods section for further details.

**Patient consent for publication** Not applicable.

**Ethics approval** The study was approved by the Swedish Ethical Review Authority DNR 2020-02491. Participants gave informed consent to participate in the study before taking part.

**Provenance and peer review** Not commissioned; externally peer reviewed.

**Data availability statement** No data are available.

**ORCID iDs**
Åse Lundin http://orcid.org/0000-0001-9980-7653
Sara Wallström http://orcid.org/0000-0001-7579-4974

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
