## [Reviewer comments · BMJ Open]

ARTICLE DETAILS

TITLE (PROVISIONAL)	Suffering out of sight but not out of mind—interpreting experiences of sick leave due to chronic pain in a community setting: a qualitative study
AUTHORS	Lundin, Åse; Ekman, Inger; Wallström, Sara; Andréll, Paulin; Lundberg, Mari

VERSION 1 – REVIEW

REVIEWER	Koechlin, Helen University Children's Hospital Zürich, Psychosomatics and Psychiatry
REVIEW RETURNED	17-Aug-2022

GENERAL COMMENTS	General comments Thank you for letting me review this interesting manuscript. This study examines the experiences of people living with chronic pain during their part- or full-time sick leave from work by means of semi-structured interviews. The main theme is “suffering out of sight but not out of mind”, describing participants’ perception of being invisible with their suffering. The manuscript reads well and the findings are very interesting and touching to read. Much of what you find is suspected or has been found in quantitative research as well, but hearing directly from people with lived experience leaves a special mark. Introduction The introduction covers the important literature for the topic. However, I believe that the study aim (on p. 6) could be a bit more specific: experiences with regard to what? Personal life, social relationships, emotional state, financial struggles, ...? It would be helpful to specify the aim(s). Methods Please include the inclusion criteria in the “participants and setting” paragraph. Table 1 belongs to the Findings section. While I appreciate that you provide the interview guide in the supplement, I think it would be helpful to give examples on questions that you used in the main manuscript. People who are not familiar with qualitative research might need a bit more information to picture this type of data collection. Further, I would also recommend to specify what you mean by “their situation” when you say that “participants were asked to narrate their situation freely”. Looking at the Supplement, the different areas
---

	("work and support", "self-efficacy and confidence in one's ability", ...) are actually quite specific. Some of the questions in the supplement are in a closed format and might be suggestive, e.g., "Being on sick leave and living with pain can also affect how you feel about your body. Do you feel you can trust your body to do what you want to do?". What was the rationale behind using this type of questions instead of more open questions? Also, it seems like this format is not quite in line with the statement you make in the Methods section, namely "participants were asked to narrate their situation freely". I would suggest you describe the interview process in more detail to avoid confusions: did you ask all questions to all participants? Were these "back-up questions" in case someone had troubles talking freely? Etc. What is NVivo? Does this need a reference? I appreciate that you contacted patient organizations. Were they also involved in the formation of the research question(s)? Findings A gap is needed between Table 2 and the title "an identity under threat". Maybe add a citation to the subtheme "Feeling constantly tired"? There is one for all other subthemes and I think they impressively undermine your interpretation. Maybe this is my lack of understanding of qualitative methodology, but I am not sure you need the paragraph "interpretation of the whole". For me, the theme of being treated justly is not as prevalent in your data, maybe with the exception of "constantly chasing recognition". If you need a theoretical framework, maybe something that relates more to identity, feeling valued in a society, etc. would make more sense. But as I mentioned, this is just a suggestion, you are the experts. Discussion I find it very interesting that employers are suggesting that people with chronic pain are "lazy" or "complainers". In a qualitative study (currently under review), we examined pain concepts of pediatricians, and unfortunately, our results pointed in a similar direction. Maybe the promotion of the biopsychosocial model and the new ICD-11 classification (especially Chronic Primary Pain) would help to decrease stigmatization? I wonder what the authors think about that, maybe this would be an interesting topic for future studies as well. One final comment: under "patient consent form" you state "not required", but in the Methods section you say that "written consent was obtained from each participant" – which one is correct?
--	---

REVIEWER	Hanson, Coral
REVIEW RETURNED	Edinburgh Napier University, School of Health and Social Care 30-Dec-2022

GENERAL COMMENTS	Thank you for the opportunity to review this interesting study, which contains some important information about the experiences of people on sick leave due to chronic pain. I think that the
---

	manuscript needs some work before it is suitable for publication. At the moment it is a little mixed up. Introduction The introduction is well written. Methods In my opinion, the methods and results are mixed up. Participants and setting This section is a mixture of method and results. Methods should describe the inclusion criteria (I can't see this), how people were recruited, what options for interviews were offered. Results should include how many were invited, how many people took part, and their characteristics (table 1). Data collection Methods should describe the type of interviews offered, how they were recorded and the interview guide. Results should include how long interviews took, what interviews were performed where, and the amount of data collected. Results. I think that the main theme and interpretation of the whole is a strong and relevant to the subject of the manuscript. However, I am not sure how strongly the lack of workplace (and healthcare professional) support is evident within the sub-themes, which are focused mainly on the personal experiences resulting from being in pain. Did the participants make suggestions about what would have helped in terms of support from treatment or their workplaces? This might help link to the idea of a 'just institution' that is introduced as part of your interpretation in the discussion. Discussion To me there is a bit of a disconnect between what I have read in the results, and the interpretation in the discussion. In addition to the point above about 'just institutions' please consider the following: Page 15 line 8 onwards – the sentences 'our findings are important because if the if the participants had learned these lessons (e.g., coping strategies) earlier, sick leave might have been shortened or prevented. The participants in our study explained that they might have continued working if their struggles had been recognized at the workplace or by the healthcare system from the start.' Please revisit your results to ensure that these points are evident. I have re-read them several times and I am not really seeing the link. Conclusion I am not sure that the manuscript does provide information to health care personnel, authorities, policymakers, and workplaces about the importance of partnership – as per my comments above. This comment also applies to the abstract conclusion.
--	--

REVIEWER	Banks, Jonathan University of Bristol, Social & Community Medicine
REVIEW RETURNED	05-Jan-2023

GENERAL COMMENTS	I have identified some issues that could be addressed in the paper P5. 30-34. I'm not sure the term 'genuine' is needed when referring to people's efforts to return to work, it's implying the existence of people who only make superficial attempts to return to work. I also think that you need to explain what you mean by
--

	adding interdisciplinary rehabilitation in brackets (the meaning is implied but not clear) Page 6 40-54. Compliant is not the right term in relation to inclusion criteria. Participants don't comply with inclusion criteria, they have characteristics that match or don't match with the inclusion criteria. If there are inclusion criteria then these need to be explicit in the paper or table. No sampling criteria are mentioned which I presume means there were none but this could be clarified if the inclusion criteria were clear. Given the reach of social media and the pain support organisations, eighteen expressions of interest seems very low – I think some comment on this (either here or in limitations would be helpful) Page 6-7, table 2. The paper doesn't tell us anything about the pain experienced by the participants - location, causes, duration, prognosis. Participants are treated as a homogenous group which is not necessarily problematic but some more details would provide important contextual detail. Page 7, 30-31. The number of pages of data is superfluous information and has no analytical relevance Page 11-12. The quote at the bottom of the page needs to be contextualised and given an introduction, it currently lacks meaning Page 13, line 10. 'they thought they were met with...' I'm not sure this is conveying what it is supposed to - I would suggest that participants were either met with accusations of being complainers etc or they thought that people perceived them as lazy etc but to say they thought that they experienced something indicates uncertainty over their own narrative which I don't think is the case here Page 15, lines 42-45. The point about a nuanced understanding being developed around participants developing a new understanding or new insight following their experiences is important to the paper but considering the emphasis given to this in the discussion I feel that it is underdeveloped in the results featuring in only one sub theme. This issue is further highlighted on page 16 lines 10-15 where the point is made that participants explained that if their issues had been recognised earlier then they may have been able to continue working – I've not found reference to this in the results section. Page 17-18, lines 53-60, 3-10. The point about the visibility of symptoms is well argued but does avoid the more difficult question of how institutions can relate to people with invisible symptoms. Those who are the gatekeepers in institutions might argue that they need some sort of assessment criteria - therein lies the challenge for us and whilst it is not the role of this paper or study to solve this some reflection on the issue might help to move the discussion forward. As a concluding comment I feel that the authors place a lot of emphasis on their interpretive methodology (phenomenological hermeneutics). I am not overly familiar with that approach but the process that is described reads very much like thematic analysis. The paper emphasises the role of interpretation, but the data and
--	---

	subsequent discussion feels descriptive. This isn't necessarily a problem, but I'm not sure where the phenomenology fits in or what it is adding.
--	---

VERSION 1 – AUTHOR RESPONSE

Reviewer: 1

Dr. Helen Koechlin, University Children's Hospital Zürich

Comments to the Author:

1. General comments

Thank you for letting me review this interesting manuscript. This study examines the experiences of people living with chronic pain during their part- or full-time sick leave from work by means of semi-structured interviews. The main theme is “suffering out of sight but not out of mind”, describing participants’ perception of being invisible with their suffering. The manuscript reads well and the findings are very interesting and touching to read. Much of what you find is suspected or has been found in quantitative research as well but hearing directly from people with lived experience leaves a special mark.

Author’s response: Thank you very much for reading and reviewing our manuscript and thank you for your supportive and helpful feedback.

2. Introduction

The introduction covers the important literature for the topic. However, I believe that the study aim (on p. 6) could be a bit more specific: experiences with regard to what? Personal life, social relationships, emotional state, financial struggles, ...? It would be helpful to specify the aim(s).

Author’s response: We agree that the aim is phrased in a general manner, but we had an intention with phrasing the aim in this way. We wanted to capture a broad picture of the lived experiences of persons being on sick leave due to chronic pain and hear the narratives from their view and explore what experiences they found had an impact on their life. Not being too specific in formulating the aim, was an aim in itself. By being general in our opening research question, “I understand you have been living in pain for a long time. Would you tell me about what this time has been like for you?” opened up for a wide range of topics within the studied phenomenon and covered diverse areas of life (such as the ones you mention; family life, relationships, emotional state, financial struggles, work etc). Which we, in line with the aim of the method used as well, believe gave varied, rich and deep insights into the multifaceted and complex phenomenon of being on sick leave due to chronic pain.

3. Methods

Please include the inclusion criteria in the “participants and setting” paragraph.

Author’s response: Thank you for your suggestion, we have now stated the inclusion criteria in the “participants and setting” section, page 5-6.

4. Table 1 belongs to the Findings section.

Author’s response: We agree with the reviewer, we have now moved table 1 to the findings section, page 8.

5. While I appreciate that you provide the interview guide in the supplement, I think it would be helpful to give examples on questions that you used in the main manuscript. People who are not familiar with qualitative research might need a bit more information to picture this type of data collection. Further, I would also recommend to specify what you mean by “their situation” when you say that “participants were asked to narrate their situation freely”. Looking at the Supplement, the different areas (“work and support”, “self-efficacy and confidence in one’s ability”, ...) are actually quite specific.

Author’s response: Thank you for your comment, we have now added examples of the questions asked into the main manuscript as well as examples of “back-up questions” used to encourage further narration, page 6. We have also rephrased the above-mentioned sentence and clarified the methods section regarding the interview process, see further reply on comment 6.

6. Some of the questions in the supplement are in a closed format and might be suggestive, e.g., “Being on sick leave and living with pain can also affect how you feel about your body. Do you feel you can trust your body to do what you want to do?”. What was the rationale behind using this type of questions instead of more open questions? Also, it seems like this format is not quite in line with the statement you make in the Methods section, namely “participants were asked to narrate their situation freely”. I would suggest you describe the interview process in more detail to avoid confusions: did you ask all questions to all participants? Were these “back-up questions” in case someone had troubles talking freely? Etc.

Author’s response: Thanks for your comment. We agree that some of the questions are more closed format. While formulating the interview guide, we chose to provide examples since our experience of interviewing has been that initiating questions this way can enhance narration, and also make it more concrete for the participants. In relation to experiences of chronic pain, prior qualitative studies have identified the close relationship between the lived body and the objective body as part of the phenomenon chronic pain (Lundberg et al, 2007; Bullington, 2013). We therefore thought it useful to include some questions including bodily experiences to making the interview questions more concrete.

We involved patient organizations in the recruitment of the participants into this study, but in hindsight we also acknowledge that we could have benefited greatly from having patient organizations and/or representatives on board with us in the development of the interview guide and its questions.

Regarding the second part of your comment, we have now restructured the text as well as described the interview process in more detail, page 6, hopefully this made the method section regarding how the interviews were performed clearer.

References:

*Lundberg M, Styf J, Bullington J. Experiences of moving with persistent pain--a qualitative study from a patient perspective. *Physiotherapy theory and practice*. 2007;23(4):199-209.*

*Bullington J. *The Expression of the Psychosomatic Body from a Phenomenological Perspective*, Dordrecht: Springer2013.*

7. What is NVivo? Does this need a reference?

Author’s response: NVivo is a frequently used CAQDAS-program (CAQDAS = Computer-assisted qualitative data analysis software). It does not perform the analysis for you, but it is a tool to work with large quantities of texts and aids in highlighting and sorting units of text into themes and subthemes. It is a software commonly used and looking at other published research using NVivo we have not seen that they give any reference to it other than the software’s name, the company’s name and the version.

We have now added the information regarding the company name and version of the program as well what type of software it is to be as clear and precise as possible in the Methods section, Analysis and interpretation of data, page 7.

8. I appreciate that you contacted patient organizations. Were they also involved in the formation of the research question(s)?

Author’s response: Patient organizations and their representatives were unfortunately not involved in the formation of research question or interview guide. In hindsight, this is something we should have done and we believe the study would have benefitted greatly from involving patient organizations at a much earlier stage and to a greater extent. These patient organizations and their channels will also be used to disseminate the study results to reach a wider audience beyond the academic community.

9. Findings

A gap is needed between Table 2 and the title “an identity under threat”.

Author’s response: A gap has been added between Table 2 and the first theme “an identity under threat”, page 9. We have noticed the tables “move” a bit depending on the version of Word used to open the manuscript etc. We hope we were able to fix this issue and that the spacing between text and table is now appropriate.

10. Maybe add a citation to the subtheme “Feeling constantly tired”? There is one for all other subthemes and I think they impressively undermine your interpretation.

Author’s response: A quote has been added under the sub-theme “feeling constantly tired”, page 12.

11. Maybe this is my lack of understanding of qualitative methodology, but I am not sure you need the paragraph “interpretation of the whole”. For me, the theme of being treated justly is not as prevalent in your data, maybe with the exception of “constantly chasing recognition”. If you need a theoretical framework, maybe something that relates more to identity, feeling valued in a society, etc. would make more sense. But as I mentioned, this is just a suggestion, you are the experts.

Author’s response: The paragraph “interpretation of the whole” is describing the third part of the analysis and is an integral part of the method used (Lindseth & Norberg, 2004). It is a part of the analysis where the themes are reflected on in relation to the research question and the context of the study, but where you also bring in the naïve understanding, our pre-understanding of the phenomenon, discussions among the authors along with literature to widen and deepen our understanding of the text (Lindseth & Norberg, 2004). It moves further and is a more abstract interpretation than just the themes from the structural analysis itself.

This part of the analysis is not a strict methodological procedure, where the literature is not regarded as a framework per se, but where we instead use our imagination as well as try to find associations with relevant literature and use the literature as a way of illuminating the text but also let the text illuminate the chosen literature (Lindseth & Norberg, 2004). We found Ricoeur’s texts about just institutions do just that, and it provided a possibility to widen our horizon and interpret and understand the text in a different light. In the subtheme “constantly chasing recognition” the connection to the literature is quite clear and straightforward, but we also found that with the other themes and subthemes the comprehensive interpretation could be enhanced by looking at them from an ethical and justice point of view, where the threat to their identities and bodies in relation to meeting the outside world exacerbated the feeling of invisibility and injustice and vice versa.

References

Lindseth A, Norberg A. A phenomenological hermeneutical method for researching lived experience. Scandinavian Journal of Caring Sciences. 2004;18(2):145-53.

12. Discussion

I find it very interesting that employers are suggesting that people with chronic pain are “lazy” or “complainers”. In a qualitative study (currently under review), we examined pain concepts of pediatricians, and unfortunately, our results pointed in a similar direction. Maybe the promotion of the biopsychosocial model and the new ICD-11 classification (especially Chronic Primary Pain) would help to decrease stigmatization? I wonder what the authors think about that, maybe this would be an interesting topic for future studies as well.

Author’s response: It is indeed very interesting, and as you have experienced and acknowledge as well, quite troubling. We really think that the new ICD-11 classification that acknowledge chronic pain as a disease in its own right is a big step in the right direction and hopefully can be part of reducing the stigma surrounding chronic pain.

In Sweden we have come some way regarding how persons suffering from chronic pain are treated within the health care system and by health care professionals compared to a few decades ago, but there is still a very long way to go. We agree that promoting the biopsychosocial model is vital when working with this group of patients and the new ICD-11 classification which for the first time contains codes for chronic pain and acknowledges it as a disease is a very important step in the right direction. According to your suggestion, we have raised this question in the discussion section, page 18-19, as well as in Conclusions, clinical implications, and future research, page 20-21 .

13. One final comment: under “patient consent form” you state “not required”, but in the Methods section you say that “written consent was obtained from each participant” – which one is correct?

Reply: Thank you very much for your question, we interpret both as correct information since they regard different aspects of consent. Written consent to take part in the study was obtained from each participant as stated in the methods section, that information is correct. Under the headline “patient consent form” in the footnotes of the article the journal submission guidelines states:

“Any article that contains personal medical information about an identifiable living individual requires the patient's explicit consent before we can publish it. We will need the patient to sign our consent form, which requires the patient to have read the article.”

We do not have any identifiable personal, medical information in the article and thus believe we do not require this journal-specific consent form from the participants regarding publishing purposes, hence the statement “not required” under this headline. It is a bit confusing to us as well, please let us know if this explanation was not sufficient.

Thank you for structured feedback and the time spent on this manuscript. Your suggestions have improved the readability of the manuscript.

Reviewer: 2

Dr. Coral Hanson, Edinburgh Napier University, Durham University
Comments to the Author:

1. Thank you for the opportunity to review this interesting study, which contains some important information about the experiences of people on sick leave due to chronic pain. I think that the manuscript needs some work before it is suitable for publication. At the moment it is a little mixed up.

Author's response: Thank you very much for your feedback and your valuable comments, which have greatly improved the manuscript. We have tried to make the manuscript more structured and clearer, especially what goes where in the method/finding's sections, please see points below.

2. Introduction

The introduction is well written.

Author's response: Thank you very much for your feedback.

3. Methods

In my opinion, the methods and results are mixed up.

Participants and setting

This section is a mixture of method and results.

Methods should describe the inclusion criteria (I can't see this), how people were recruited, what options for interviews were offered. Results should include how many were invited, how many people took part, and their characteristics (table 1).

Data collection

Methods should describe the type of interviews offered, how they were recorded and the interview guide. Results should include how long interviews took, what interviews were performed where, and the amount of data collected.

Author's response: In order to present the method and results in a more logical way, we have now made several revisions and restructured both the method and findings section and hopefully the information is now easier to follow. Table 1 has been moved to the findings section, page 8. We have also added and clarified the inclusion criteria in the methods section, page 5-6.

4. Results.

I think that the main theme and interpretation of the whole is a strong and relevant to the subject of the manuscript. However, I am not sure how strongly the lack of workplace (and healthcare professional) support is evident within the sub-themes, which are focused mainly on the personal experiences resulting from being in pain. Did the participants make suggestions about what would have helped in terms of support from treatment or their workplaces? This might help link to the idea of a 'just institution' that is introduced as part of your interpretation in the discussion.

Author's response: The interviews contained very rich and varied data and to be able to make the data justice and shine enough light on the participants' multifaceted experiences we decided to divide the result into two different studies. Thus, not all domains included in the interview guide is covered by this article, which is focused on the personal experiences of suffering from chronic pain while on sick leave.

5. Discussion

To me there is a bit of a disconnect between what I have read in the results, and the interpretation in the discussion. In addition to the point above about 'just institutions' please consider the following:

Page 15 line 8 onwards – the sentences 'our findings are important because if the if the participants had learned these lessons (e.g., coping strategies) earlier, sick leave might have been shortened or prevented. The participants in our study explained that they might have continued working if their struggles had been recognized at the workplace or by the healthcare system from the start.' Please revisit your results to ensure that these points are evident. I have re-read them several times and I am not really seeing the link.

Author's response: Thanks for giving concrete and specific feedback. Since the results and findings in this study (or in any phenomenological hermeneutical research) is not just the structural analysis. i.e., the themes and subthemes but also the interpretation of the whole, the comprehensive understanding of the text. This means not all the results discussed are found explicitly in the structural analysis part, but in our overall comprehension. At the same time, we do acknowledge your comment and that there is a bit of disconnection between the result section and the discussion section, especially regarding the section mentioned in your comment. We have revisited the text and the analysis process and we do find that we have these points in the data, but not explained fully in text regarding the themes. We have now clarified this in the manuscript, also with a quote from one of the participants on page 13, and hopefully the link between the result and the subsequent discussion is now more evident.

6. Conclusion

I am not sure that the manuscript does provide information to health care personnel, authorities, policymakers, and workplaces about the importance of partnership – as per my comments above. This comment also applies to the abstract conclusion.

Author's response: Thank you very much for your comment. We agree, we have taken all your valuable comments into consideration and the conclusion is not formulated well and accurately. We have revisited the analysis, the findings and the discussion and reformulated the conclusion, page 3 and page 20-21.

Reviewer: 3

Dr. Jonathan Banks, University of Bristol
Comments to the Author:

I have identified some issues that could be addressed in the paper

Author's response: Thank you very much for your feedback and your valuable comments, which have greatly improved the manuscript. We have addressed the issues below.

1. P5. 30-34. I'm not sure the term 'genuine' is needed when referring to people's efforts to return to work, it's implying the existence of people who only make superficial attempts to return to work. I also think that you need to explain what you mean by adding interdisciplinary rehabilitation in brackets (the meaning is implied but not clear)

Author's response: We agree, the term genuine is not needed and is not correctly used in this context. We have removed the use of that word. We have also clarified and rephrased the sentence regarding interdisciplinary rehabilitation as an effort aiming at helping people with chronic pain in returning to work, page 4.

2. Page 6 40-54. Compliant is not the right term in relation to inclusion criteria. Participants don't comply with inclusion criteria, they have characteristics that match or don't match with the inclusion criteria. If there are inclusion criteria then these need to be explicit in the paper or table. No sampling criteria are mentioned which I presume means there were none but this could be clarified if the inclusion criteria were clear.

Given the reach of social media and the pain support organizations, eighteen expressions of interest seem very low – I think some comment on this (either here or in limitations would be helpful)

Author's response: According to your comment we have now addressed this issue and removed the term compliant and instead added and clarified the inclusion criteria in the methods section, page 5-6.

Indeed, patient organizations have a wide reach. We underestimated the outreach. On the first day of Facebook post by one of the patient organizations we quickly received 10-12 answers from that post, and subsequently asked the organization to remove the post early as we by then started to get close to the target sample size (where phenomenological research often aim to involve 5 to 25 persons who have experienced the phenomenon) (Polkinghorne, 1989).

References

Polkinghorne DE. *Phenomenological research methods. Existential-phenomenological perspectives in psychology: Exploring the breadth of human experience.* New York, NY, US: Plenum Press; 1989. p. 41-60.

3. Page 6-7, table 2. The paper doesn't tell us anything about the pain experienced by the participants - location, causes, duration, prognosis. Participants are treated as a homogenous group which is not necessarily problematic but some more details would provide important contextual detail.

Author's response: Thank you very much for your suggestion, we have now added characteristics and details regarding location and duration of pain amongst the participants, into table 1 as well as in text below the table, page 8.

4. Page 7, 30-31. The number of pages of data is superfluous information and has no analytical relevance

Author's response: Thank you very much for your comment, we agree and have now removed this information from the manuscript.

5. Page 11-12. The quote at the bottom of the page needs to be contextualised and given an introduction, it currently lacks meaning

Author's response: Thank you very much for your suggestion, we have now contextualized the quote and given it a short introduction in the preceding text, we hope this made the quote more clearly related to the sub-theme, page 12.

6. Page 13, line 10. 'they thought they were met with...' I'm not sure this is conveying what it is supposed to - I would suggest that participants were either met with accusations of being complainers etc or they thought that people perceived them as lazy etc but to say they thought that they experienced something indicates uncertainty over their own narrative which I don't think is the case here

Author's response: Thank you very much for your comment, we agree that this segment has been phrased oddly and have now revised the sentence to clarify the meaning, page 13. Thank you for pointing this out.

7. Page 15, lines 42-45. The point about a nuanced understanding being developed around participants developing a new understanding or new insight following their experiences is important to the paper but considering the emphasis given to this in the discussion I feel that it is underdeveloped in the results featuring in only one sub theme. This issue is further highlighted on page 16 lines 10-15 where the point is made that participants explained that if their issues had been recognised earlier then they may have been able to continue working – I've not found reference to this in the results section.

Author's response: Being one subtheme, it is also inherent in the theme, as well as the overall interpretation of the text, where we interpret that the lack of institutional support enforces the invisibility and the suffering, and with this support in place their sick leave might have been shortened. The reason we chose to emphasize this subtheme in the discussion is that this part of the findings is what we found had not been uncovered in earlier studies thus providing new insights of interest do discuss further. We agree that there is a bit of disconnection between the result section and the discussion section. We have revisited the text and the analysis process and we do find that we have these points in the data, but not explained fully in text regarding the themes. We have now clarified this in the manuscript, also with a quote from one of the participants on page 13, and hopefully the link between the result and the subsequent discussion is now more evident.

8. Page 17-18, lines 53-60, 3-10. The point about the visibility of symptoms is well argued but does avoid the more difficult question of how institutions can relate to people with invisible symptoms. Those who are the gatekeepers in institutions might argue that they need some

sort of assessment criteria - therein lies the challenge for us and whilst it is not the role of this paper or study to solve this some reflection on the issue might help to move the discussion forward.

Author's response: The invisibility of symptoms (also symptoms beyond pain, such as fatigue) is a large and important question and as this study imply, also a barrier these persons face when encountering different institutions in society. It is a great challenge indeed and we think that it is a bit out of scope to elaborate further into this theme within the word limit and scope of this article.

9. As a concluding comment I feel that the authors place a lot of emphasis on their interpretive methodology (phenomenological hermeneutics). I am not overly familiar with that approach but the process that is described reads very much like thematic analysis. The paper emphasizes the role of interpretation, but the data and subsequent discussion feels descriptive. This isn't necessarily a problem, but I'm not sure where the phenomenology fits in or what it is adding.

Author's response: Thank you very much for your comment. The aim of this study was both to describe and to interpret the experiences of persons being on sick leave due to chronic pain. Methodologically it is a thematic analysis of sorts, thematic analysis is sometimes considered an umbrella term for qualitative analyses with the aim to examine themes or patterns of meaning within data, which is what we have done in this study using phenomenological hermeneutics. But using this method we also try to abstract the text and interpret what the essential meaning of the text is.

We move between the manifest and the latent content, were the more concrete structural analysis and exemplified with quotes comes close to the text and therefore can be seen as more descriptive, to the comprehensive interpretation of the text where we instead try to decipher what the text is really trying to tell us, beyond the words. This is where the interpretation becomes vital and where phenomenological hermeneutical investigation is about the meaning of lived experience. We look at all the subthemes, themes, our naïve understanding along with discussion among the authors and associated literature to try to go further than descriptive level and, in this study, gain broader and deeper insights into the essential meaning of the lived experiences of chronic pain and sick leave, through the use of interpretation. We have made some small changes to the section "interpretation of the whole", page 15, where we hope this section now more clearly conveys our interpretation and the essential meaning of the studied phenomenon.

VERSION 2 – REVIEW

REVIEWER	Koechlin, Helen University Children's Hospital Zürich, Psychosomatics and Psychiatry
REVIEW RETURNED	16-Mar-2023

GENERAL COMMENTS	General Thank you for letting me review the revised version of the manuscript. I can see that the authors put a lot of effort into the revision, and I greatly appreciate their willingness to receive and implement the feedback of all reviewers. I have some small comments for authors' consideration. Introduction In my opinion, the introduction is much stronger now. I specifically like how authors differentiate between the descriptive and the interpretative levels of qualitative work, I think this is helpful for the understanding of your methodology. I have nothing to add. Methods I appreciate the more detailed description of the use of the interview guide.
---

	Results I think your Results section is clear and structured, and I appreciate the additional quote you added. One very small comment: I struggle with the wording on p. 14, l. 26-27 (“The participants’ experiences of living with pain were not all undesirable in that some had learned important life lessons.”) – what you report afterwards sounds more like coping strategies for pain self-management and less like “life lessons”. Consider re-phrasing as this might sound somewhat patronising. While I can see that the newly added sentences on p. 13 are helpful, I would consider re-phrasing the first part of the first sentence: “With pain being experienced as a hidden and questionable illness (...)”. This perception is from the outside, isn’t it? For patients themselves, pain is neither hidden nor questionable. Maybe you can phrase this in a way that clearly indicates that this is an outside perception, not an internal experience. Discussion I would be careful with statements such as “They were insights and understanding that would not have been reached without the participants facing the process of illness and sick leave.” (p. 16) – this can sound as if you are minimizing participants’ experiences, and we simply cannot know whether participants would have reached the same conclusions while being healthy and fully able to work.
--	--

REVIEWER	Hanson, Coral Edinburgh Napier University, School of Health and Social Care
REVIEW RETURNED	21-Feb-2023

GENERAL COMMENTS	Thank you for the opportunity to review this revised manuscript, which is now better structured and easier to read. I have a few very minor comments. Methods and results section structure: The restructuring of these sections meant that they are now much clearer and easy to follow. One page 7, line 186, you may want to add who into the first sentence: Out of the 18 people who contacted the first author Links between results and discussion: The authors have added in data to the results section that now makes the discussion more meaningfully related to the results. Given that the authors have stated in their response that they are planning a further paper focusing on other aspects of the data, is it rewording the sentence to the data analysis section ‘All parts of the text were considered, but the parts not relevant to the research question were not included in the analysis.’ to make this clearer? Page 18, line 449 – should this read managing in the following sentence: Another benefit is that this new classification may potentially shift the focus from finding a cause to instead manage the impact of the chronic pain for the patient, for example through multimodal treatments Conclusion in manuscript and abstract: These now contain only elements reported in the study. General comment: As this is manuscript is not written in the authors first language, there are a few small grammatical errors or phrasing of sentences that can be a little odd. I am sure that these will be picked up during the editing process.
---

VERSION 2 – AUTHOR RESPONSE

Reviewer: 2

Dr. Coral Hanson, Edinburgh Napier University, Durham University

Comments to the Author:

1. Thank you for the opportunity to review this revised manuscript, which is now better structured and easier to read. I have a few very minor comments.

Author's response: Thank you very much for reading and reviewing our revised manuscript and thank you for your supportive and helpful feedback through this process. We have addressed your comments below.

2. Methods and results section structure:

The restructuring of these sections meant that they are now much clearer and easy to follow.

One page 7, line 186, you may want to add who into the first sentence: Out of the 18 people who contacted the first author

Author's response: Thank you for your feedback and also your suggestion, we agree and have added the word "who" into the sentence, thank you for pointing this out as it makes much more sense now.

3. Links between results and discussion:

The authors have added in data to the results section that now makes the discussion more meaningfully related to the results. Given that the authors have stated in their response that they are planning a further paper focusing on other aspects of the data, is it rewording the sentence to the data analysis section 'All parts of the text were considered, but the parts not relevant to the research question were not included in the analysis.' to make this clearer?

Author's response: Thank you for your comment. We have rephrased the text slightly, see page 7. We have also added information regarding further analyses and studies on the material, see page 20.

4. Page 18, line 449 – should this read managing in the following sentence: Another benefit is that this new classification may potentially shift the focus from finding a cause to instead manage the impact of the chronic pain for the patient, for example through multimodal treatments

Author's response: Thank you for your comment, we agree and have now changed from manage to managing in the mentioned sentence according to your suggestion, see page 19.

5. Conclusion in manuscript and abstract:

These now contain only elements reported in the study.

Author's response: Thank you very much for your feedback.

6. General comment:

As this manuscript is not written in the authors' first language, there are a few small grammatical errors or phrasing of sentences that can be a little odd. I am sure that these will be picked up during the editing process.

Author's response: Thank you for your feedback, you are correct and English is not the first language of any of the authors. We have now completed a second professional proofread and corrected spelling, grammar and phrasing errors in the manuscript. As there were many small grammatical changes throughout the manuscript, they have not all been marked independently.

Reviewer: 1

Dr. Helen Koechlin, University Children's Hospital Zürich

Comments to the Author:

1. General

Thank you for letting me review the revised version of the manuscript. I can see that the authors put a lot of effort into the revision, and I greatly appreciate their willingness to receive and implement the feedback of all reviewers. I have some small comments for authors' consideration.

Author's response: Thank you very much for reading and reviewing our revised manuscript and thank you for your supportive and helpful feedback through this process. We have addressed your comments below.

2. Introduction

In my opinion, the introduction is much stronger now. I specifically like how authors differentiate between the descriptive and the interpretative levels of qualitative work, I think this is helpful for the understanding of your methodology. I have nothing to add.

Author's response: Thank you very much for your feedback.

3. Methods

I appreciate the more detailed description of the use of the interview guide.

Author's response: Thank you very much for your feedback.

4. Results

I think your Results section is clear and structured, and I appreciate the additional quote you added. One very small comment: I struggle with the wording on p. 14, l. 26-27 (“The participants’ experiences of living with pain were not all undesirable in that some had learned important life lessons.”) – what you report afterwards sounds more like coping strategies for pain self-management and less like “life lessons”. Consider re-phrasing as this might sound somewhat patronising.

Author’s response: Thank you for your feedback. We have rephrased the sentence slightly to avoid patronizing language, see page 15.

5. While I can see that the newly added sentences on p. 13 are helpful, I would consider re-phrasing the first part of the first sentence: “With pain being experienced as a hidden and questionable illness (...)”. This perception is from the outside, isn’t it? For patients themselves, pain is neither hidden nor questionable. Maybe you can phrase this in a way that clearly indicates that this is an outside perception, not an internal experience.

Author’s response: Thank you for your feedback. Several of the participants described how they experienced pain as a hidden disease, but as you state we also made the interpretation that this perception is (at least affected) from the outside. It is important to remember that for the participants themselves the pain was very tangible and visible. We have made a small alteration of the phrasing to make both aspects clearer, see page 14 and again on page 16.

6. Discussion

I would be careful with statements such as “They were insights and understanding that would not have been reached without the participants facing the process of illness and sick leave.” (p. 16) – this can sound as if you are minimizing participants’ experiences, and we simply cannot know whether participants would have reached the same conclusions while being healthy and fully able to work.

Author’s response: Thank you for your suggestion. We agree with your comment, both regarding the risk of minimizing their experiences but also that we cannot draw such conclusions because we just don’t know. We have removed that short statement and rephrased the sentence by focusing on the implication and importance of that finding, see page 17.